# Correlation between maternal and umbilical cord 25-hydroxy-vitamin D levels over a range of values. A prospective observational study from the United Arab Emirates

Magnus Jutell[1]☉, Shakura Bhat[2]‡, Maria Lindstrom Bagge[2]‡, Per-Erik Isberg[3]☉, Nana Wiberg [ORCID][3,4,5]☉ *

**1** Center for Psychiatry, Amager Hospital, Copenhagen, Denmark, **2** Women and Children, Sheikh Khalifa Hospital, Ajman, United Arab Emirates, **3** Department of Statistics, Lund University, Lund, Sweden, **4** Department of Clinical Sciences, Lund University, Malmö, Sweden, **5** Department of Gynecology and Obstetrics, Sjaelland University Hospital, Roskilde, Denmark

☉ These authors contributed equally to this work.
‡ SB and MLB also contributed equally to this work.
* nana.wiberg@med.lu.se

**Data Availability Statement:** Data cannot be shared publicly because of a high confidentiality in the United Arab emirates. The data underlying the

## Abstract

Worldwide vitamin D insufficiency is remarkably prevalent in both children and adults, including pregnant women. The total amount of the vitamin is best measured by 25-hydroxy-vitamin D (25(OH)D), which is a measurement of total serum cholecalciferol 25 (OH)D3 and ergocalciferol 25(OH)D2. There is a known correlation between maternal and umbilical cord blood (UCB) 25(OH)D; however, whether specific maternal demographics or comorbidities influence the correlation remains uncertain. This prospective observational study was designed to study if maternal 25(OH)D levels, maternal age and BMI, amount of supplementation, mode of delivery, diabetes, hypertension/preeclampsia, or sunlight exposure had an impact on the correlation. Women were enrolled in the study at admission to the labor ward. If they agreed to participate, venous blood was directly collected and analyzed for 25(OH)D. The UCB was sampled after delivery from the unclamped cord and immediately analyzed for 25(OH)D. ANOVA, Fisher's exact test, Pearson's correlation, and test of the differences between correlations using Fisher's z-transformation with Bonferroni correction were used accordingly. Of the 298 women enrolled, blood from both the mother and umbilical cord was analyzed successfully for 25(OH)D in 235 cases. The crude correlation between maternal and UCB 25(OH)D was very strong over all values of 25(OH)D (r = 0.905, $R^2$ = 0.821, p <0,001) and remained strong independently of maternal demographics or comorbidities (r ≥ 0.803, $R^2$ ≥ 0.644, p <0.001). For women who delivered by caesarean section in second stage the correlation was strong (r ≥ 0.633, $R^2$ ≥ 0.4, p <0.037). Test of differences between correlations showed significant stronger correlation in women with unknown 25(OH)D3 supplementation compared to women receiving 10.000 IU/week (p = 0.02) and 20.000IU/week (p = 0.01) and that the correlation was significantly stronger for women with a BMI of 25–29.9 compared to women with a BMI of <24.9 (p = 0.004) and 30–34.9 (p = 0.002). 213 (91%) women had lower 25(OH)D compared to the neonate, with a mean

results presented in the study are available from
vidya.Jakapure@skmca.ae.

**Funding:** We received no funding for the study.

difference of -13.7nmol/L (SD = 15.6). In summary, the correlation between maternal and UCB 25(OH)D is very strong throughout low to high maternal levels of 25(OH)D with lower levels in maternal blood. Typical maternal demographics and comorbidities did not affect the transition.

## Introduction

Vitamin D is a pleiotropic secosteroid hormone, historically known for its role in calcium and phosphate metabolism, with deficiencies leading to rickets and osteomalacia. Equally important, data have shown that vitamin D is linked to numerous essential functions, such as modulation of the immune and neuromuscular systems, the metabolism of glucose, and cell growth [1–3].

Despite the challenging and ongoing discussions on how to measure vitamin D, current guidelines recommend measuring total 25(OH)D, the most abundant, stable and biologically inactive form of the vitamin measuring cumulative $25(OH)D_2$ and $25(OH)D_3$ [4]. The hydroxylation of 25(OH)D into its active metabolite 1,25-dihydroxy vitamin D 1,25(OH)2D, and the further catabolism into less active metabolites is regulated by a feedback loop involving two main enzymes CYP27B1 and CYP24A1. These are during pregnancy uncoupled, resulting in elevated and fluctuating maternal concentrations of circulating $1,25(OH)D_2$ [5]. Maternal 1,25 $(OH)D_2$ is thought not to pass the placenta. Fascinating new and upcoming data from epigenetic studies shows that placental functionality and epigenetics dictate the fetal supply of 25 (OH)D's active metabolite, why it is remarkable that many authors still refer to passive transport [6–8]. Simple descriptive studies show a moderate to strong correlation between maternal and UCB 25(OH)D to no correlation if the mother is deficient and the ratio is seldom described as negative. This can either be due to a complex pathway between the mother and the fetus or, more simply, the design of the studies [7, 9–11]. There are studies suggesting that maternal diabetes, risk for preeclampsia, delivery of a small for gestational age neonate, preterm rupture of membranes, and dystocia are associated with low vitamin D but according to the Cochrane review from 2019 more studies are needed to prove the evidence for any of these associations [12–14]. Having in mind that the placental function is highly influenced by both maternal BMI, preeclampsia, and diabetes it is surprising that the correlation between maternal and UCB 25(OH)D never have been described for those subgroups.

The threshold for determining 25(OH)D deficiency in both adults and neonates remains a subject of ongoing debate. The most recognized cut-off values for adults and pregnant women indicating deficiency are 25(OH)D $\leq$50nmol/l and for neonates $\leq$30nmol/L [15, 16]. However, it has been argued that the ideal level in adults should be >75 nmol/L [5, 15]. In a study from the United Arab Emirates, up to 69% of pregnant women had 25(OH)D < 30nmol/L, and the awareness of the importance of having normal levels of vitamin D was low, also in well-educated women [17].

This study aimed to investigate the correlation between maternal and UCB levels of 25 (OH)D at delivery and to study if specific maternal demographics or comorbidities influenced the correlation.

## Materials and methods

This prospective observational study was conducted from 27[th] September until 31[st] December 2021 at Sheikh Khalifa Medical City, Women and Children, Ajman, United Arab Emirates, with approximately 1300 deliveries per year, in collaboration with Lund University, Lund, Sweden.

## Procedure

During the study period, all women coming to the labor ward at Sheikh Khalifa Medical City, Women and Children, were verbally informed about the study by the coordinating midwife in charge. If the woman wanted to participate, she signed the informed consent, allowing the researchers to sample blood from her and the umbilical cord, enter her medical record after discharge, and handle the questionnaire. If the father attended the delivery, he signed the informed consent with the mother. In contrast, if the father was not attending, informed consent was taken after delivery, permitting the researchers to enter and use data from the neonatal medical records. Inclusion criteria were admission for delivery and a signed written informed consent. Women who did not speak Arabic, English, or Urdu, were excluded from the study. As per routine, maternal venous blood was collected for a standard panel of blood analyses, and 25(OH)D was included if she was enrolled. Immediately after birth, i.e., after vaginal delivery or cesarean section, the UCB was sampled from all neonates before the delayed cord clamping procedure. The blood was analyzed directly for acid-base and electrolyte values by the stationary blood gas analyzer in the labor ward and for 25(OH)D in the central laboratory. If both the maternal and UCB samples were successfully analyzed, we regarded the samples as paired and complete. If the UCB sample was hemolyzed, which is very common due the high viscosity of cord blood, we still collected and described maternal demographics and comorbidities. On the other hand, if the analysis of maternal blood failed, neonatal data were collected.

Before discharge, the mothers answered a questionnaire in writing. If the mother were illiterate, she would answer the questionnaire in the presence of research officials who would record her answers in writing. She was asked about her education, physical activity, daily time spent outside, intake of vitamin D (for ladies coming to our hospital for prenatal care, Euro D, Cholecalciferol 10.000 IU, Euro-Pharm, Canada, was the vitamin prescribed), the importance of compliance, and her plan of supplementation of vitamin D to the baby and breastfeeding (not all data was processed for this article). Basic maternal and obstetric, as well as neonatal data were obtained, from the electronic medical records (Cerner, Missouri, USA), and entered directly into a specific study database.

## Biochemical analysis

Throughout the day (24/7), maternal blood and UCB were sent directly after sampling to the hospital's central laboratory for serum 25(OH)D analysis by an electrochemiluminescence binding assay using LIAISON® 25 OH Vitamin D TOTAL (dynamic range 10–374 nmol/L. The inter- assay and intra-assay CV were 3.2% and 8.3%, respectively). The laboratory is internationally certified by Analytics Professional (CAP) and accredited by *The International Organization for Standardization (ISO)*. UCB acid-base values, including electrolytes, hemoglobin, and hematocrit, were analyzed directly by a stationary blood gas analyzer (ABL90, Radiometer, Copenhagen, Denmark). Maternal vitamin D deficiency was defined as 25(OH)D <50nmol/L. The cut-off for severe deficiency is often set between 20-30nmol/L, why we decided to use 25 (OH)D <25nmol/L as reference. The criteria for neonatal deficiency was 25(OH)D<30nmol/L (Fig 1) [15, 16].

## Statistical analysis

One-way ANOVA, Fisher's exact test, test of difference between correlations using Fisher's z-transformation with Bonferroni correction of p-values and Pearson's correlation were used when appropriate. Descriptive statistics were presented as mean with standard derivation (SD) and medians with interquartile range (IQR). All tests were two-tailed, and a P-value < 0.05

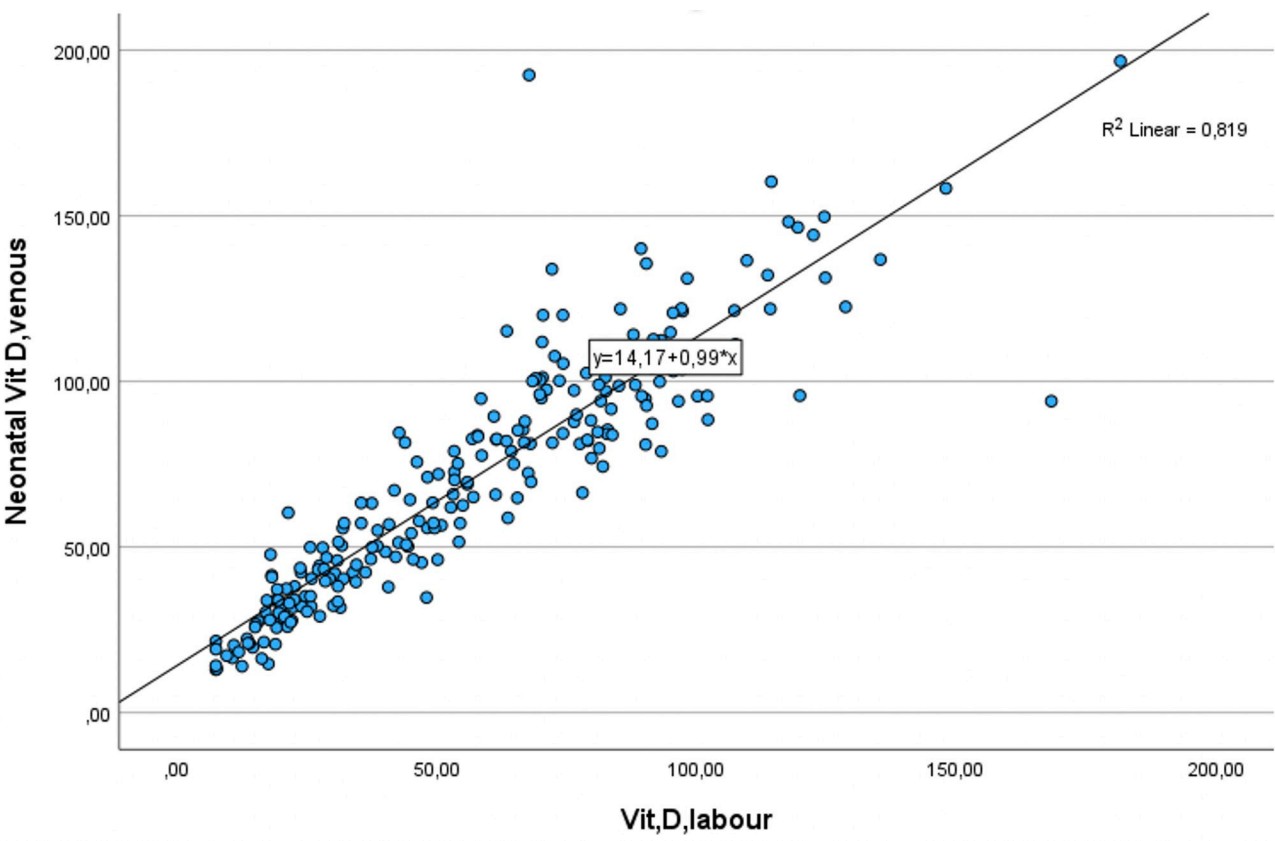

**Fig 1. Scatter plot of maternal and UCB serum 25(OH)D mmol/L.**

was considered statistically significant. Analysis by IBM SPSS Statistics for Windows, Version 29.0. Armonk, NY: IBM Corp.

### Ethics

The study was approved well before the start, by the Ministry of Health and Prevention, United Arab Emirates (the committee is an established IRB with the jurisdiction to provide ethical oversight and ethical approval on human subject research) with diary number: MOHAP/DXB-REC/SSS/No. 79/2021. All participants signed an informed consent before enrollment.

### Results

Of the 320 women coming to the labor ward for delivery during the study period, 298 women were eligible and enrolled in the study. 235 participants had complete maternal/UCB samples, of those 110 (46.8%) were diagnosed with 25(OH)D deficiency, whereas only 33 (14%) of their neonates met the criteria. If comparing the whole cohort to the cohort with complete maternal/UCB 25(OH)D values, there was only very small and clinically irrelevant differences (data not shown).

Among subgroups of different maternal 25(OH)D levels (severe deficiency <25nmol/L (n = 64), moderate deficiency 25–49.9nmol/l (n = 72), normal status ≥50nmol/L (n = 150)), there was no significant difference in maternal age, gestational age at delivery, placenta weight, fetal weight, level of UCB ionized calcium (Ca2+ mmol/L), nationality, daily time spent outside, prevalence of diabetes, delivery mode or Apgar scores ≤ 7 after 5 minutes, as seen in

**Table 1. Maternal and neonate characteristics divided according to different levels of maternal 25-hydroxy-vitamin-D (25(OH)D) at delivery.** SD: standard derivation, IQR: interquartile range, BMI; body mass index, IU; international units, AS; Apgar score, [†] One-way ANOVA [‡] Fishers exact test.

| | | *Group I* <br> **Maternal 25(OH)D <25nmol/L** <br> **n = 64** | *Group II* <br> **Maternal 25(OH)D 25nmol/L to < 50nmol/L.** <br> **n = 72** | *Group III* <br> **Maternal 25(OH) D > 50nmol/L** <br> **n = 150** | *P* value |
|---|---|---|---|---|---|
| | | **Mean ± SD** <br> **Median (IQR)** | **Mean ± SD** <br> **Median (IQR)** | **Mean ± SD** <br> **Median (IQR)** | |
| **Maternal age** | | 29.8 ± 6.25 <br> 28 (10) | 30.2 ± 5.24 <br> 30 (8) | 30.9 ± 6.22 <br> 31 (9) | 0.425[†] |
| **Gestational week at delivery** | | 38.1 ± 1.9 <br> 38 (3) | 38.4 ± 3.1 <br> 39 (2) | 38.2 ± 2.2 <br> 39 (2) | 0.795[†] |
| **Maternal level of 25(OH)D (nmol/L)** | | 17.5 ± 4.9 <br> 18.9 (7.2) | 37.3 ± 7.55 <br> 37.4 (13.2) | 81.7 ± 23.73 <br> 77.8 (28.5) | 0.001[†] |
| **Maternal BMI (kg/m2)** | | 29.9 ± 5.8 <br> 29.0 (7.5) | 31.8 ± 5.8 <br> 31.0 (7.6) | 29.0 ± 6.2 <br> 29.1 (7.6) | 0.006[†] |
| **Placenta weight (g)** | | 486 ± 137 <br> 500 (213) | 532 ± 167 <br> 500 (200) | 536 ± 113 <br> 500 (100) | 0.457 |
| **Neonate weight (g)** | | 3068 ± 582 <br> 3098 (633) | 3073 ± 696 <br> 3170 (640) | 3038 ± 541 <br> 3110 (628) | 0.009[†] |
| **Umbilical cord blood 25(OH)D (nmol/L)** | | 28.1 ± 9.6 <br> 27.9 (13.1) | 49.4 ± 12.3 <br> 48.5 (15.9) | 97.9 ± 26.3 <br> 94.9 (30.7) | 0.001[†] |
| **Umbilical cord blood Ca$^2$+ (mmol/L)** | | 1.40 ± 0.09 <br> 1.41 (0.13) | 1.40 ± 0.11 <br> 1.42 (0.13) | 1.41 ± 0.09 <br> 1.42 (0.12) | 0.610[†] |
| | | n (%) | n (%) | n (%) | |
| **Nationality** | Middle east <br> Africa <br> Southeast Asia | 36 (56.3) <br> 8 (12.5) <br> 20 (31.3) | 38 (52.8) <br> 17 (23.6) <br> 17 (23.6) | 90 (60.0) <br> 24 (16.0) <br> 36 (24.0) | 0.400[‡] |
| **Daily time-spent-outside** | None <br> 0–30 min <br> 30–60 min <br> > one hour | 38 (66.7) <br> 18 (31.6) <br> 1 (1.8) <br> 0 | 36 (59.0) <br> 22 (36.1) <br> 3 (4.9) <br> 0 | 73 (58.4) <br> 49 (39.2) <br> 1 (0.8) <br> 2 (1.6) | 0.455[‡] |
| **Maternal Diabetes** | No <br> Yes | 52 (81.3) <br> 12 (18.8) | 52 (72.2) <br> 20 (27.8) | 100 (66.7) <br> 50 (33.3) | 0.092[‡] |
| **Maternal hypertension/ preeclampsia** | No <br> Yes | 53 (82.8) <br> 11 (17.2) | 60 (84.5) <br> 11 (15.5) | 138 (94.5) <br> 8 (5.5) | 0.010[‡] |
| **Delivery mode** | Vaginal delivery <br> Cesarean section | 48 (76.2) <br> 15 (23.8) | 47 (65.3) <br> 25 (34.7) | 99 (67.7) <br> 52 (34.7) | 0.283[‡] |
| **AS ≤ 7 at 5 min** | | 1 (1,6) | 2 (2.8) | 2 (1.4) | NA |

Table 1. In contrast the BMI was higher in the group with 25-50mmol/L 25(OH)D but only compared to the group with normal levels of 25(OH)D. Also, there were more women with preeclampsia/hypertension with lower 25(OH)D levels.

There was no significant correlation between placental weight or UCB-hematocrit to UCB-25(OH)D level (r = 191, *p* = 0.188; and r = -0.061, *p* = 0.369, respectively) and no neonate was born with hypocalcemia defined as ionized calcium <1.10mmol/L. The mean ionized calcium levels in UCB for mothers without/with supplementation were 1.44mmol/L ± 0.06; and 1.41 mmol/L ± 0.07, respectively (*p* = 0.136) (Fig 2).

Table 2 shows the correlation coefficients, the coefficients of determination, and test of difference of correlation. All correlations were significant and very strong, except delivery by caesarean section in the second stage of labor which showed a weaker probability (*p* = 0.037) compared to the other subgroups and a strong correlation (r = 0.633). When testing the

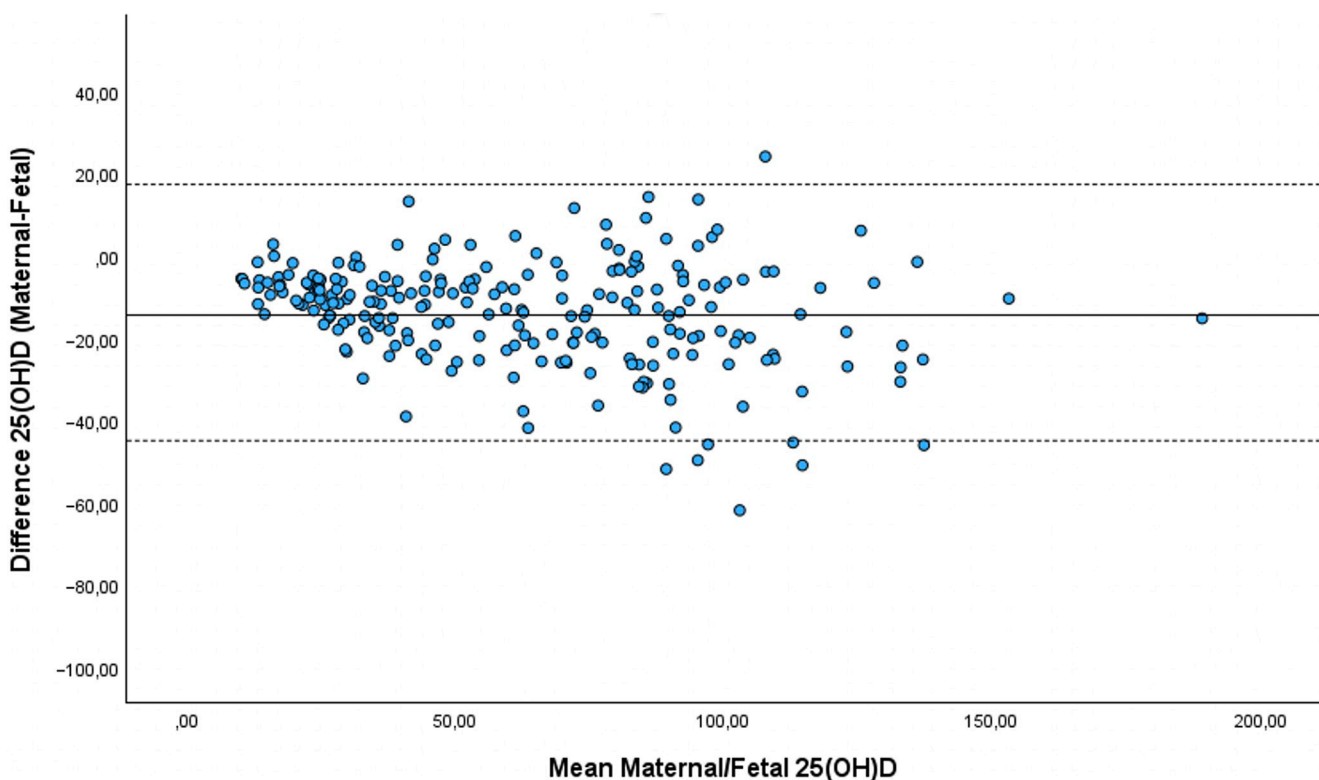

**Fig 2. Maternal and mean maternal-fetal serum 25(OH)D levels presented with ±2SD.** The mean difference between maternal and UCB serum 25(OH)D was negative (higher levels in the UCB compared to the maternal blood) in 91% of cases (-13.69nmol/L, SD = 15.56). This difference was not significantly related to maternal 25(OH)D level, placental weight, or delivery mode ($p > 0.493$).

difference between correlations in the respective subgroups we found significant differences within subgroups of 25(OH)D3 supplementation during pregnancy and maternal BMI. The serum 25(OH)D in women with unknown supplementation dosage showed a stronger correlation to UCB 25(OH)D than women supplemented with 10.000 IU/week (z-score -2.931, $p = 0.02$), and 20.000IU/week (z-score -3.105, $p = 0.01$). Within the subgroups of maternal BMI, the correlation was stronger for women with a BMI 25–29.9kg/m$^2$ compared to women with a BMI of <24.9kg/m$^2$ (z-score -3.419, $p = 0.004$), and 30–34.9kg/m$^2$ (z-score 3.587, $p = 0.002$). In all other subgroups of maternal characteristics, except the ones mentioned above, the differences in correlations were insignificant (data not shown).

## Discussion

In areas with a high prevalence of vitamin D insufficiency, particularly in low-income communities, this study posits that a deeper understanding of the maternal-neonatal correlation of 25 (OH)D is needed. While the link between maternal 25(OH)D levels and maternal comorbidities remains a subject of debate, we found it intriguing to explore whether these suggested associations could impact the correlation between maternal and UCB 25(OH)D levels since this mapping may provide even more guidance in estimating the risk of neonatal 25(OH)D insufficiency, mainly when only the mother's 25(OH)D levels are available.

There is an alleged association between maternal 25(OH)D levels and pathologies such as obesity, diabetes, hypertension, dystocia and small-for-gestational-age-neonates as described and discussed by multiple publications [11, 14, 18–21]. Accessible vitamin D is essential for

**Table 2. Correlation between maternal and UCB 25(OH)D levels and test of difference between correlations using Fisher's z-transformation with Bonferroni correction of p-values (only significant results shown).** IU: International Units, CS: Cesarean Section, BMI: Body Mass index kg/m$^2$, ‡10.000IU/week–Unknown, ‡‡2x10.000IU/week–Unknown, * BMI (<24.9)–(25–29.9), ** BMI (25–29.9)–(30–34.9).

| | | n | r | R$^2$ | p | *Test of difference between correlation p* |
|---|---|---|---|---|---|---|
| All cases | | 235 | 0.905 | 0.821 | < 0.001 | |
| **Maternal level of 25(OH)D** | <25nmol/L | 53 | 0.696 | 0.485 | < 0.001 | |
| | 25–49.5nmol/L | 57 | 0.583 | 0.340 | < 0.001 | |
| | ≥50nmol/L | 125 | 0.697 | 0.486 | < 0.001 | |
| **Vit-D supplementation** | None | 17 | 0.952 | 0.907 | < 0.001 | 0.02‡ |
| | 10.000IU/week | 73 | 0.853 | 0.728 | < 0.001 | 0.01‡‡ |
| | 2x10.000IU/week | 37 | 0.803 | 0.644 | < 0.001 | |
| | Unknown | 106 | 0.938 | 0.880 | < 0.001 | |
| **Delivery mode** | Unassisted vaginal delivery | 175 | 0.922 | 0.85 | < 0.001 | |
| | Elective | 18 | 0.908 | 0.825 | < 0.001 | |
| | CS first stage | 24 | 0.945 | 0.893 | < 0.001 | |
| | CS second stage | 11 | 0.633 | 0.4 | 0.037 | |
| **Diabetes** | No | 171 | 0.915 | 0.838 | < 0.001 | |
| | Yes | 64 | 0.875 | 0.765 | < 0.001 | |
| **Hypertension/preeclampsia** | No | 210 | 0.903 | 0.816 | < 0.001 | |
| | Yes | 21 | 0.928 | 0.861 | <0.001 | |
| **BMI** | < 24.9 | 53 | 0.863 | 0.745 | < 0.001 | 0.004* |
| | 25–29.9 | 69 | 0.960 | 0.922 | < 0.001 | 0.002** |
| | 30–34.9 | 64 | 0.864 | 0.747 | < 0.001 | |
| | >35 | 47 | 0.925 | 0.856 | < 0.001 | |
| **Daily time spent outside** | None | 125 | 0.894 | 0.799 | <0.001 | |
| | 0–30 min | 71 | 0.922 | 0.85 | <0.001 | |
| | 30–60 min | 4 | 0.998 | 0.997 | 0.002 | |

The crude correlation coefficient between maternal and UCB 25(OH)D was 0.905 (95% CI 0.88–0.93) ($p < 0.001$), R$^2$ = 0.821.

fetal development and post-uterine life health; however, it is uncertain to which degree the maternal level impacts the fetal level and if there are subgroups where the correlation is altered.

We observed a strong correlation across varying levels of maternal 25(OH)D, maintaining a consistent maternal/UCB transition despite the presence of common maternal comorbidities, also in caesarean section in the second stage of labor but with a lower probability. Only 11 women underwent caesarean section in the second stage, leading us to hypothesize that this contributed to the observed decrease of probability. Our analysis of delivery modes revealed no significant differences in correlation within this subgroup (data not shown). Additional research is warranted to explore whether different modes of delivery, particularly caesarean section at various stages of labor, impact the maternal/UCB correlation of 25(OH)D.

Notably, even in cases of preeclampsia, the correlation between maternal and UCB levels of 25(OH)D remained unaltered. This observation is particularly intriguing given that, during pregnancy, the placenta plays a crucial role in converting inactive 25(OH)D3 into its active metabolite, 1,25(OH)2D3, and releasing it into both the maternal and umbilical circulations [6]. A recent study from Sweden have shown that women with 25(OH)D deficiency in early pregnancy presented with higher risk of delivery before week 34 or delivering a neonate who was small for gestational age [22]. The Vitamin D receptor and 1alpha-hydroxylase, responsible for converting 25(OH)D to 1,25(OH)2D3, exhibit significantly heightened expression in the first trimester compared to the third trimester. Additionally, the Vitamin D receptor is upregulated in preeclamptic placentas compared to those from healthy pregnancies providing possible insights on the role played by 25(OH)D and 1,25(OH)2D3 in the formation and physiology of the placenta [23, 24], which supports our findings.

Successfully, we obtained a high number of paired maternal and UCB 25(OH)D samples within a short time frame of hours, and contradictory to most studies, we found the 25(OH)D levels to be higher in UCB. One possible explanation for the observed inconsistency in the correlation between maternal and UCB 25(OH)D levels is the absence of heterogeneity in the timing of maternal 25(OH)D measurements across the published studies. The time span of maternal measurement of 25(OH)D varied from weeks to months before delivery, and the known fluctuation of different vitamin D metabolites during pregnancy further underscores the importance of precise timing in measurements [8, 9, 11, 12, 25–31]. Another reason could be the composition of UCB known with higher hemoconcentration, but we did not find a correlation between UCB 25(OH)D and UCB hematocrit. In this context, it is essential to mention that vitamin D and the mean corpuscular hemoglobin and volume are significantly associated [32].

Although the maternal level of 25(OH)D is thought to increase simultaneously and corresponding to oral intake of vitamin D, there is growing evidence for an individual response due to differences in the absorption dependent on our genetics/epigenetics, why it is speculated if some require a higher dose than others to be sufficiently substituted [33, 34]. However, results from regions with similar latitudes has shown a significant correlation between UCB and maternal $25(OH)D_3$ in un-supplemented women whereas in a predominantly white population from a northern latitude the main predictor for UCB 25(OH)D was delivery in the summer months [10, 35]. Our results suggested that women with unknown supplementation had significantly stronger correlation of maternal and UCB 25(OH)D compared to women with documented supplementation dosage but due to the expected heterogenicity of the unknown group it is difficult to draw any further conclusions regarding supplementation and maternal/UCB correlation of 25(OH)D other than it stayed strong regardless of supplementation.

A correlation has been established between 25(OH)D deficiency and elevated BMI, with evidence also suggesting that thin women exhibit a more favorable response to vitamin D supplementation compared to their overweight counterparts [36, 37]. Within the different groups of maternal BMI we found a significant difference in the correlation where the correlation were stronger if the mother were overweight compared to normal weight or obese and that the mean maternal BMI were highest in the group of maternal serum 25(OH)D of 25 - <50nmol/L. Our findings partially deviate from prior publications, as the observed differences in BMI and 25(OH)D status were statistically significant, albeit exceedingly small. This minimal discrepancy poses a challenge in deriving conclusive interpretations.

Emerging data indicate that genetic factors play a crucial role in the transport and metabolism of vitamin D and its metabolites across the placental barrier but many authors are still referring to the passive transport of 25(OH)D over the placenta [6]. A metanalysis from 2022 investigated the relationship between both maternal and fetal gene variations (Single nucleotide polymorphisms (SNPs)) and UCB 25(OH)D levels and found that common genetic variations are associated with individual levels of 25(OH)D in UCB [7]. Vitamin D is introduced to organ cells after endocytosis and is either bound to vitamin-D-binding protein or is albumin-bound [38–40]. A study from 2022 investigating placental metabolism and uptake of 25(OH)D hypothesized a similar process in the placenta and demonstrated that the term-placenta actively uptakes 25(OH)D3 by endocytosis, and in the presence of albumin, they saw a significant increase in the expression of CYP24A1 gene, associated to the regulation of the amount of vitamin D in the body. 25(OH)D3 was metabolized by the placenta into (24,25(OH)2D3 and 1,25(OH)2D3) and released into the maternal and fetal circulation, but varied degrees of influence were observed, dependent upon the specific trimester of the placenta and decidua [23, 41]. Also, the exposure to vitamin D was found to be associated with a rapid effect on a set of messenger RNA expressed by the placenta. This exposure, together with the individual's genome, seems to edict the transcriptional response to vitamin D [6]. Our results suggested

that UCB 25(OH)D was higher than the mothers in 91% of the cases implying that other factors than just the maternal transition of 25(OH)D and the respective metabolites influence the UCB level with possible genetic factors and placental physiology.

Our discovery that neonatal 25(OH)D levels, as described above, are higher in 91% of our study subjects, coupled with the fact that blood samples were collected within hours before delivery, offers valuable insights into the maternal-neonatal 25(OH)D relationship at birth. This knowledge can assist clinicians, especially those in resource-constrained settings, in accurately assessing the risk of neonatal 25(OH)D insufficiency. Consequently, it enables them to effectively prioritize screening and preventive measures, such as those for rickets.

Already in 1984 Hollis et. al. showed, in a small study consisting of 12 white and 10 colored mothers, that racial factors seemed to play a role in both maternal and fetal vitamin D status [42]. In this study we choose to stratify mothers based on nationality since the skin phenotype is a subjective assessment if not based on dermatological tests and definitions. Between the three groups defined in our study, the majority of women came from the middle east, where most women are covered leading to very limited exposure to sunlight which could explain the insignificance between the subgroups of 25(OH)D levels and a steady correlation.

We analyzed total 25(OH)D ($25(OH)D_2 + 25(OH)D_3$). The different metabolites of vitamin D, have various stability and biological activity, mainly explained by the affinity to the vitamin D receptor [4]. Although new arrays for measuring the different vitamin D metabolites, the international recommendation is still to measure the cumulative level by quantifying the amount of 25(OH)D which is stable also up to 5 years of storage at -80 degrees Celsius [4, 43]. Total 25(OH)D has a long half-life-time giving only minor variations within short periods, and it shows a significant response to dermal vitamin production (increase $25(OH)D_3$) and to a lesser degree also to supplementation intake [44].

Our prospective observational study indicates a strong correlation between maternal and UCB 25(OH)D across the investigated factors. Furthermore, our findings suggest that maternal BMI and supplementation dosage may exert an influence on this correlation. However, it is essential to interpret this conclusion with caution, given the heterogeneity within supplementation practices and the subtle variations in BMI observed in our study.

**Strength**: The study was conducted in a high endemic region and with a high number of paired maternal and umbilical cord blood samples taken within a specific and narrow time frame. Data was not retrieved from a register but directly from the EMR, avoiding methodological errors.

**Weakness:** Due to hemolysis of the UCB, it was impossible to analyze 25(OH)D in 66 cases.

## Conclusion

There is a very strong correlation between maternal and UBC 25(OH)D levels with higher levels in UCB. The correlation remained strong after categorization into different maternal comorbidities, exposure to sunlight, delivery mode and vitamin D supplementation. Significant differences in correlation were observed within supplementation dose and maternal BMI but should be interpreted with caution. Based on new knowledge, the correlation between the mother and the UCB seems to be dictated by epigenetics and immunomodulatory factors and is not only based on maternal 25(OH)D levels and passive transport of 25(OH)D over the placenta. Future studies are warranted to map the epigenetic landscape in women with comorbidities to offer the best vitamin D deficiency treatment during pregnancy.

## Acknowledgments

We want to thank the women and their partners for participating in the study and to our colleagues (doctors, midwives, and nurses) for their enthusiasm in helping to conduct the study.

## Author Contributions

**Conceptualization:** Magnus Jutell, Nana Wiberg.

**Data curation:** Magnus Jutell, Shakura Bhat, Maria Lindstrom Bagge, Nana Wiberg.

**Formal analysis:** Magnus Jutell, Per-Erik Isberg, Nana Wiberg.

**Investigation:** Magnus Jutell, Shakura Bhat, Per-Erik Isberg, Nana Wiberg.

**Methodology:** Shakura Bhat, Maria Lindstrom Bagge, Per-Erik Isberg, Nana Wiberg.

**Project administration:** Magnus Jutell, Maria Lindstrom Bagge, Nana Wiberg.

**Resources:** Nana Wiberg.

**Software:** Per-Erik Isberg.

**Supervision:** Nana Wiberg.

**Validation:** Shakura Bhat, Nana Wiberg.

**Visualization:** Per-Erik Isberg, Nana Wiberg.

**Writing – original draft:** Magnus Jutell, Shakura Bhat, Per-Erik Isberg, Nana Wiberg.

**Writing – review & editing:** Magnus Jutell, Shakura Bhat, Maria Lindstrom Bagge, Per-Erik Isberg, Nana Wiberg.

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
