## [Decision Letter · Decision Letter 0]

11 Apr 2023

PONE-D-23-01799Excellent correlation between maternal and fetal 25(OH)D at delivery. A prospective observational study from the United Arab Emirates.PLOS ONE

Dear Dr. Wiberg,

Thank you for submitting your manuscript to PLOS ONE. After careful consideration, we feel that it has merit but does not fully meet PLOS ONE’s publication criteria as it currently stands. Therefore, we invite you to submit a revised version of the manuscript that addresses the points raised during the review process.

We look forward to receiving your revised manuscript.

Kind regards,

Sanjoy Kumer Dey, M.D

Academic Editor

PLOS ONE

Journal Requirements:

"No 

only funding was payment for the analysis"

"NO"

Reviewers' comments:

Reviewer's Responses to Questions

**Comments to the Author**

1. Is the manuscript technically sound, and do the data support the conclusions?

Reviewer #1: Partly

Reviewer #2: No

Reviewer #3: Partly

2. Has the statistical analysis been performed appropriately and rigorously? 

Reviewer #1: I Don't Know

Reviewer #2: I Don't Know

Reviewer #3: Yes

3. Have the authors made all data underlying the findings in their manuscript fully available?

Reviewer #1: No

Reviewer #2: Yes

Reviewer #3: Yes

4. Is the manuscript presented in an intelligible fashion and written in standard English?

Reviewer #1: No

Reviewer #2: Yes

Reviewer #3: Yes

5. Review Comments to the Author

Reviewer #1: Wiberg et al have assessed the associations between maternal and cord 25(OH)D. Although this is interesting in an Arab population, the findings are not novel. The authors would benefit from performing a more extansive literature review to see that such an association has been reported many times previously, and referring to some of this literature.

Abstract

I would not typically use the word "excellent" to describe a correlation. Please reword in the title and elsewhere.

Please use . for decimal place marker rather than ,

25(OH)D typically stands for 25-hydroxyvitamin D, not "vitamin D". Please correct in abstract and introduction. Please ensure all abbreviations are defined within abstract.

The aims of the study are not entirely clear from the abstract - the background in the abstract discusses the unknown role of the placenta but this establishing this does not seem to be the aim of the study. Please provide a clear aim and the background should be relevant to this.

"Of note, delivery by acute cesarean section in the second stage affected the correlation (r=0.644, p=0.024)". What is meant by "affected"? This is not scientific terminology.

" Of the 303 women, 138 (47.7%) had vitamin D deficiency (defined as 25(OH)D<50nmol/L), whereas only 34 (13.8%) of UCB samples fulfilled the criteria for neonatal deficiency (25(OH)D<30nmol/L)." There were only 237 women with maternal and UCB samples so it is impossible to compare the two percentages of deficiency as it may not reflect the same set of women. Also why was a different cut-off used for maternal and fetal defiency?

Overall the results in the abstract would benefit from some consideration of what is relevant to present here as at the moment it feels like presentation of a random selection of results. This may reflect the lack of clear aims/objectives in the abstract however.

There is no conclusion to the abstract.

Background

" During pregnancy, the vitamin is also thought to facilitate the transport of several nutrients across the placenta, including calcium (4)" Please list exactly which nutrients vitamin D facilitates the transport of across the placenta, and support with some primary evidence rather than a review article.

" Low vitamin D has been associated with increased risk for preeclampsia, small for gestational age fetuses, preterm rupture of membranes, and dystocia, but according to WHO, the evidence is low and more trials are warranted (5–7)." The authors reference here review and guidelines from 2007-2012. Please undertake a more up to date literature review with regards to this as there have been numerous observational studies and trial data reported in the last 10-15 years.

Ref 8 is incompletely report.

"It is a general assumption that maternal levels primarily control fetal levels and that in women with vitamin D deficiency, the fetus will suffer from insufficient supply." I would strongly disagree that this is an assumption, there are many published studies showing a strong correlation between maternal and UCB 25(OH)D.

I would suggest review of 10.1210/clinem/dgac263 which shows maternal and offspring genetic variation in the vitamin D metabolism pathway affects UCB 25(OH)D.

"The definition of hypovitaminosis in pregnancy is 25(OH)D below 50nmol/l and in neonates below 30nmol/L." Please provide a reference for this. Please also note that many guidelines and societies suggest different values to this for defining vitamin D deficiency. Please review.

The background does not give clear justification for the aims of the study. Also the primary aim of establishing the correlation between maternal and UCB 25(OH)D has been reported previously on many occasions. What is novel about the aims of this work?

MEthods

What is EMR?

Was is a interviewer directed questionnaire or a written questionnaire completed by the participant?

obstetrical is not a word

Was all data normally distributed? If not, what statistical methods was used?

From where was maternal anthropometric/BMI data obtained

How was GDM/PET/PIH diagnosed?

Results

Were those without complete maternal/UCB 25OHD similar to those with that data?

Why are all data in table 1 presented as mean (SD) and median (IQR). If normally distributed present as mean (SD), if not, present only as median (IQR).

What are the units for cord Ca2+

Why is BMI presented stratified rather than as mean (SD) /median (IQR)?

The data shown in table 1 suggested BMI did differ by 25(OH)D status, contrary to the statement in the text.

I would suggest rather than looking at % GDM/Caesarean by 25(OH)D grouping, that you examine 25(OH)D as a continuous variable by GDM y/n, LSCS y/n, PET y/n etc

What does AS<7 mean?

"Subdivision into the different levels of maternal 25(OH)D showed a consistent correlation in all three groups without a poorer correlation with low maternal values, Fig 1." This sentence does not make sense. It cannot be consistent and poorer. Please provide the r values.

"Also, we could not show a correlation between UCB142 Ca2+ and vitamin D supplementation dose (n = 141)" - I would not expect this to be a linear correlation. Was neonatla hypocalcaemia more common in infants of mothers who did not use any supplementation?

Discussion

I disagree with the comment that the correlation between maternal and UCB 25(OH)D is rarely described. I would suggest the authors undertake a further literature review on this. Maghbooli BMC Pregnancy 2007, Grant Pediatrics 2013, Song Public Health Nutr 2013 for starters.

" With maternal levels <50nmol/L no newborn was born with levels under 40.5nmol/L." This sentence does not agree with what is shown in your figure.

" The evidence for an association between low 25(OH)D in the mother and complications during pregnancy such as obesity, diabetes, preeclampsia and dystocia are for discussion accordingly to the latest statement from WHO." This sentence does not make sense, but it is also important to realise that the statement the authors refer to is over 10 years old and the body of literature on this topic has advanced substantially since that time.

It is not clear to me what the relevance of a correlation between maternal and UCB 25(OH)D is stratified by these outcomes. What is more relevant is how the clinical outcome is related to maternal 25(OH)D as this would potentially offer a therapeutic intervention to prevent the outcome and/or reason to stress supplementation in that group to prevent neonatal vitamin D deficiency and associated hypocalcaemia.

Further thought should be given to the weaknesses of the study. why did 25% not have paired samples for example? Did those included represent the local population?

Of note, the order of authors on the entered data differs from that on the manuscript.

Reviewer #2: am unclear about the originality of the study. These data have been reproduced previously in similar regions.

Despite this UVB variations, data on supplementation and robustness of assay measurement are major flaws of the study. A detailed questionnaire of maternal skintype, nutritional intake of vitamin D ,sartorial habits, sunscreen use and many othe factors are not included in the analysis. Methods of sampling are not described in detail.

Reviewer #3: The authors compare total vitamin D levels in 237 blood samples out of 303 women/newborn in a cohort with high frequency of deficiency. They find, as others, a correlation between mother and child levels with slightly lower levels in women.

I have some issues that needs to be addressed before publication

Title:

Superlatives in the title is unwarranted. I would suggest “Correlation between maternal and…..”.

“Fetal” is usually an unborn offspring that develops and grows inside the uterus, as umbilical cord is investigated, I would suggest the use the term “newborn”.

The cohort:

It is stated that all women coming for delivery were invited to participate, but how many declined or were excluded (i.e. is the cohort representative?).

Instead of stratificating the women in nationality I would have preferred accord to skin color and/or tradition with covered clothing.

To me it is a bit unclear when the maternal blood sample was performed. In the abstract it is stated “at the time of admission”, however no further details are provided. Are these hours, days before delivery?

General:

A bit more discussion regarding previous findings regard vitamin D transfer through placenta is warranted. (Haddad JG Jr, Boisseau V, Avioli LV. Placental transfer of vitamin D3 and 25-hydroxycholecalciferol in the rat. J Lab Clin Med. 1971 Jun;77(6):908-15; Gupta, M. M., Kuppuswamy, G., & Subramanian, A. R. (1982). Transplacental transfer of 25-hydroxy-cholecalciferol. Postgraduate medical journal, 58(681), 408–410; Kiely M, O'Donovan SM, Kenny LC, Hourihane JO, Irvine AD, Murray DM. Vitamin D metabolite concentrations in umbilical cord blood serum and associations with clinical characteristics in a large prospective mother-infant cohort in Ireland. J Steroid Biochem Mol Biol. 2017 Mar;167:162-168 and especially all the work by Dr Jane Cleal from the University of Southampton).

Furthermore, I need some more stringency in the nomenclature between vitamin D2 (ergocalciferol) and vitamin D3 (cholecalciferol). In this context, were the supplementation I D2 or D3 form?

6. PLOS authors have the option to publish the peer review history of their article (what does this mean?). If published, this will include your full peer review and any attached files.

Reviewer #1: No

Reviewer #2: No

Reviewer #3: **Yes: **PB Szecsi

---

## [Author Response · Author response to Decision Letter 0]

16 Jun 2023

Dears, we are very grateful for the valuable comments and have worked with each of them to improve the quality of our publication. Please find the answers in the letter *comments to reviewers'

---

## [Decision Letter · Decision Letter 1]

18 Sep 2023

PONE-D-23-01799R1Correlation between maternal and fetal 25(OH)D at delivery. A prospective observational study from the United Arab Emirates.PLOS ONE

Dear Dr. Wiberg,

Thank you for submitting your manuscript to PLOS ONE. After careful consideration, we feel that it has merit but does not fully meet PLOS ONE’s publication criteria as it currently stands. Therefore, we invite you to submit a revised version of the manuscript that addresses the points raised during the review process.

We look forward to receiving your revised manuscript.

Kind regards,

Sanjoy Kumer Dey, M.D

Academic Editor

PLOS ONE

Reviewers' comments:

Reviewer's Responses to Questions

**Comments to the Author**

1. If the authors have adequately addressed your comments raised in a previous round of review and you feel that this manuscript is now acceptable for publication, you may indicate that here to bypass the “Comments to the Author” section, enter your conflict of interest statement in the “Confidential to Editor” section, and submit your "Accept" recommendation.

Reviewer #1: (No Response)

Reviewer #2: All comments have been addressed

Reviewer #3: (No Response)

2. Is the manuscript technically sound, and do the data support the conclusions?

Reviewer #1: Partly

Reviewer #2: No

Reviewer #3: Yes

3. Has the statistical analysis been performed appropriately and rigorously? 

Reviewer #1: No

Reviewer #2: Yes

Reviewer #3: I Don't Know

4. Have the authors made all data underlying the findings in their manuscript fully available?

Reviewer #1: No

Reviewer #2: Yes

Reviewer #3: No

5. Is the manuscript presented in an intelligible fashion and written in standard English?

Reviewer #1: No

Reviewer #2: No

Reviewer #3: No

6. Review Comments to the Author

Reviewer #1: Thank you for submitting the revised manuscript. I remained concerned by the originality of this work, and do feel there is some "clutching at straws" to utilise some data. I remain uncertain as to how looking at the correlations stratified by maternal demographics is actually useful unless you are perhaps suggesting that this may influence the pathway of maternal-fetal 25D transfer. If this is the case, then some discussion of this and hypothesis around how the factors would influence this would actually make the manuscript far clearer. Furthermore, to be able to state there was no difference between the correlation, a statistical correlation needs to be made between them. I would suggest discussions with a statistician as to how to achieve this.

* As per my previous comments, please rephrase the term "excellent correlation" in the abstract and elsewhere in the paper.

* "Except for acute cesarean section in the second stage

(p=0.024), the correlation remained strong after dividing the women into subgroups of

supplementation intake, BMI, diabetes, hypertension/preeclampsia, and time spent

outside (p<0.001)." I do not understand this sentence and what the authors are trying to show/what analysis has been done or why.

* Overall I find the abstract very difficult to follow. It seems that the authors have run a large number of analyses of different things - maternal/UCB correlations, possibly then across different subgroups, maternal 25D with different outcomes, and then chosen the significant factors to report. Please rewrite the abstract giving clear and exact aims, and address these in order within the abstract. eg "Of those with paired

samples, 111 (46.8%) women had vitamin D deficiency (defined as

25(OH)D<50nmol/L), whereas only 32 (13.5%) of UCB samples fulfilled the criteria for

neonatal deficiency (25(OH)D<30nmol/L). Low maternal 25(OH)D was associated with

significantly more women having hypertension/preeclampsia." Why is this all presented when the abstract states the aim is "This prospective observational study aimed to correlate the maternal 25(OH)D levels to UCB levels in a

high-risk population and to show if certain factors influenced the correlation".

Introduction

"Accordingly to a study from 2012 there is only limited amount of data when

52 it comes to the evidence for the levels and the correlations between maternal and neonatal

53 25(OH)D" I dont understand the first part of this sentence. Levels of what?

* It would be helpful to add some comment as to why you think any of these maternal factors would influence the correlation, and what the value in knowing this information would be.

Results

* Please add units of measurement to table 1

* Please review the numbers in () for HT/PET in table 1 - i do not think these are correct, and assuming these represent the % of women, do not add up to 100

* "In contrast there was a trend towards more women with preeclampsia/hypertension with lower 25(OH)D levels," It is not a trend towards, there were more women with PET/HT in those with lower 25D

* "The correlation was strong, and with no differences in correlation coefficients for the different subgroups of maternal 25(OH)D levels." Have you performed a statistical test to be able to categorically state there is no difference between these correlations? I would recommend discussion with a statistician. This should be done for all the subgroup analyses that have been undertaken. I do not think the conclusions that have been made that these subgroups did not affect the correlation can be made until this has been performed.

* "The scientists behind the meta-analysis..." the findings dicussed are not from the meta-analysis but from a different paper, as referenced.

* I disagree that normality can be assumed based on the n number. This is not always the case, and even independent of n, some data will not be normally distributed. The descriptive data should be presented as either mean (SD) if normal on visual inspection and median (IQR) if not normal on visual inspection.

Reviewer #2: I do not fell that this study has high standards for publication since similar data have been previously reported

Reviewer #3: Most of the issues has been clarified, however, some still needs to be addressed.

Generally, the manuscript could benefit from a native English speaking review, I can recommend American Journal Editors (AJE). Alternative, a careful proof reading would make the manuscript more readable.

Title: Abbreviation 25(OH)D is not feasible in the title (unexplained).

Affiliation: Usually department, institution, city, country is provided, this manuscript lacks some city and one report zip code. Uniformity would be prudent.

Short title: The nomenclature regarding fetal, newborn, neonatal is still not uniform (ex: in the short title “fetal” is used), as well as other places I the text.

Abstract:

Abbreviations in the abstract should be explained at firsts occurrence (25(OH)D), it is first explained at line 54.

Is p-values below zero? (p<0,000), write p<0.001 (use punctuation instead of comma). Also, present in table 2.

What is “paired samples”?, is not all comparison on paired (mother/child) samples, or is groups compared?

Material and Methods:

The abbreviation UAE should be explained, but as it is not used frequently, just write United Arab Emirates.

Correct “after vaginal and cesarean section” to “after vaginal or cesarean section”.

I presume that the vitamin D analyses were performed on a Liaison XL instrument with the LIAISON® 25 OH Vitamin D TOTAL Assay?

Table:

IQR is the interquartile range (Q3-Q1), a single cipher. However, two ciphers are presented, I assume it is Q1 and Q3. Correct IQR to first and third quartile or correct the ciphers.

Figure 1&2:

Please modify the designation of the axis, fetal/neonate, vitD labour, at least make them uniform.

SPSS is cited correctly as: IBM SPSS Statistics for Windows, Version 28.0. Armonk, NY: IBM Corp

7. PLOS authors have the option to publish the peer review history of their article (what does this mean?). If published, this will include your full peer review and any attached files.

Reviewer #1: No

Reviewer #2: No

Reviewer #3: **Yes: **Pal Bela Szecsi

---

## [Author Response · Author response to Decision Letter 1]

7 Dec 2023

We express our gratitude for your invaluable input and remarks. Please see changes in the attached documents.

---

## [Decision Letter · Decision Letter 2]

19 Dec 2023

PONE-D-23-01799R2Correlation between maternal and umbilical cord 25-hydroxy-vitamin D over a range of values. A prospective observational study from the United Arab Emirates.PLOS ONE

Dear Dr. Wiberg,

Thank you for submitting your manuscript to PLOS ONE. After careful consideration, we feel that it has merit but does not fully meet PLOS ONE’s publication criteria as it currently stands. Therefore, we invite you to submit a revised version of the manuscript that addresses the points raised during the review process.

We look forward to receiving your revised manuscript.

Kind regards,

Sanjoy Kumer Dey, M.D

Academic Editor

PLOS ONE

Journal Requirements:

Reviewers' comments:

Reviewer's Responses to Questions

**Comments to the Author**

1. If the authors have adequately addressed your comments raised in a previous round of review and you feel that this manuscript is now acceptable for publication, you may indicate that here to bypass the “Comments to the Author” section, enter your conflict of interest statement in the “Confidential to Editor” section, and submit your "Accept" recommendation.

Reviewer #1: (No Response)

Reviewer #3: All comments have been addressed

2. Is the manuscript technically sound, and do the data support the conclusions?

Reviewer #1: Partly

Reviewer #3: Yes

3. Has the statistical analysis been performed appropriately and rigorously? 

Reviewer #1: Yes

Reviewer #3: I Don't Know

4. Have the authors made all data underlying the findings in their manuscript fully available?

Reviewer #1: No

Reviewer #3: No

5. Is the manuscript presented in an intelligible fashion and written in standard English?

Reviewer #1: No

Reviewer #3: Yes

6. Review Comments to the Author

Reviewer #1: The manuscript is much improved and more clear to follow. There are some minor points below regarding the main text but the abstract does require further modification.

Abstract:

-" This prospective observational study aimed to study the correlation between maternal and UCB 25(OH)D levels in a population with high risk for 25(OH)D deficiency, secondary, to show if maternal demographics or comorbidities certain factors influenced the correlation." Please review this sentence and ensure it makes sense. I am unclear why the word secondary is included.

- Similarly, this sentence does not make sense in English "At admission for labor, respectively,

after delivery, 25(OH)D was analyzed in maternal serum and in UCB"

- I remain unclear as to what "analyzing for different maternal demographics means" - do you mean you undertook stratified analysis by these characteristics or you looked for a statistical interaction?

- Line 43 - a p value of 0.037 is significnat. Please explain why this was not significant?

- Line 44-45 - what are the actual r values for these correlations? Just giving a p value is meaningless.

-Methods Line 135 - the use of respectively in this sentence is incorrect. This is true in other places in the manuscript also.

-Results Line 162 - should be "were diagnosed"

- The text in paragraph lines 167 needs to refer to the table

- Why is the data in Table 1 presented as both mean(SD ) and median (IQR) - please prevent the relevant statistics depending on the distribution of the data. Similarly please refer to the figures in the relevant text.

- I am really not sure of the point of comparing the correlations of vitamin D supplementation to women with "unknown supplementation". Please justify as to me this seems meaningless given the potentially very mixed nature of this group.

- line 220 - a p value of 0.037 is typically considered statistically signficiant although i agree we should not a slave to the p value. Authors have also stated "All tests were two-tailed, and a P-value < 0.05 was

152 considered statistically significant.". PLease review this and the discussion accordingly.

line 275 do you mean UCB 25(OH)D here - UCB needs to be collected immediately so it is unlikely to lead to heterogenous time interval between maternal and offspring blood sampling.

-lines 334 - do you think that maternal haemodilution around the time of delivery could be contributing to the relatively lower 25OHD in the mother compared to infant. Additionally how many women received iv fluids during delivery which could have contributed to a dilutional effect?

Reviewer #3: The manuscript has improved, however, it still is hard to read (especially the abstract).

I would have used other phrases such as total instead of "cumulative vitamin D2 & D3".

Plasma, serum and blood are mentioned as source of measurement. Is that correct?

The interesting discussion is a bit speculative based upon the minor difference in mother/offspring vitamin D levels.

7. PLOS authors have the option to publish the peer review history of their article (what does this mean?). If published, this will include your full peer review and any attached files.

Reviewer #1: No

Reviewer #3: **Yes: **Pal Bela Szecsi

---

## [Author Response · Author response to Decision Letter 2]

28 Jan 2024

Thank you for the third round of revision! We are grateful for your input and hope that you will find the changes appropriate, and the manuscript suitable for publication. Please see the new document Response to reviewers.

---

## [Editor Report · Decision Letter 3]

16 Feb 2024

Correlation between maternal and umbilical cord 25-hydroxy-vitamin D levels over a range of values. A prospective observational study from the United Arab Emirates.

PONE-D-23-01799R3

Dear Dr. Nana Wiberg

We’re pleased to inform you that your manuscript has been judged scientifically suitable for publication and will be formally accepted for publication once it meets all outstanding technical requirements.

Kind regards,

Sanjoy Kumer Dey, M.D

Academic Editor

PLOS ONE
---

## [Editor Report · Acceptance letter]

26 Feb 2024

PONE-D-23-01799R3 

PLOS ONE

Dear Dr. Wiberg, 

I'm pleased to inform you that your manuscript has been deemed suitable for publication in PLOS ONE. Congratulations! Your manuscript is now being handed over to our production team.

Kind regards, 

on behalf of

Dr. Sanjoy Kumer Dey 

Academic Editor

PLOS ONE